# Flavonoids Activation of the Transcription Factor Nrf2 as a Hypothesis Approach for the Prevention and Modulation of SARS-CoV-2 Infection Severity

**DOI:** 10.3390/antiox9080659

**Published:** 2020-07-24

**Authors:** Patricia Mendonca, Karam F. A. Soliman

**Affiliations:** College of Pharmacy and Pharmaceutical Sciences, Florida A&M University, Tallahassee, FL 32307, USA; patricia.mendonca@famu.edu

**Keywords:** ACE2 receptors, COVID-19, EGCG, flavonoids, Nrf2, SARS-CoV2, thymoquinone, Vitamin D3

## Abstract

The Nrf2-Keap1-ARE pathway is the principal regulator of antioxidant and phase II detoxification genes. Its activation increases the expression of antioxidant and cytoprotective proteins, protecting cells against infections. Nrf2 modulates virus-induced oxidative stress, ROS generation, and disease pathogenesis, which are vital in the viral life cycle. During respiratory viral infections, such as the severe acute respiratory syndrome coronavirus 2 (SARS-CoV-2), an inflammatory process, and oxidative stress of the epithelium lining cells activate the transcription factor Nrf2, which protects cells from oxidative stress and inflammation. Nrf2 reduces angiotensin-converting enzyme 2 (ACE2) receptors expression in respiratory epithelial cells. SARS-CoV2 has a high affinity for ACE2 that works as receptors for coronavirus surface spike glycoprotein, facilitating viral entry. Disease severity may also be modulated by pre-existing conditions, such as impaired immune response, obesity, and age, where decreased level of Nrf2 is a common feature. Consequently, Nrf2 activators may increase Nrf2 levels and enhance antiviral mediators’ expression, which could initiate an “antiviral state”, priming cells against viral infection. Therefore, this hypothesis paper describes the use of flavonoid supplements combined with vitamin D3 to activate Nrf2, which may be a potential target to prevent and/or decrease SARS-CoV-2 infection severity, reducing oxidative stress and inflammation, enhancing innate immunity, and downregulating ACE2 receptors.

## 1. Introduction

The novel coronavirus infectious disease (COVID-19), caused by the severe acute respiratory syndrome coronavirus 2 (SARS-CoV-2), was first reported in December 2019 in Wuhan, China, and since then has emerged as a major pandemic [1,2]. The total global numbers are 14,774,887 infected and 611,599 deaths, presenting 3,858,686 total cases and 141,426 deaths only in the US, as of July 20, 2020. SARS-CoV-2 is one of the seven coronaviruses identified to cause human respiratory disease. Four of them are known to cause general cold symptoms, and the other two cause severe acute respiratory syndrome coronavirus (SARS-CoV) and the Middle East respiratory syndrome coronavirus (MERS-CoV), which killed thousands due to fatal respiratory disease [3,4]. Even though many therapies including antiviral drugs, corticosteroid therapy, drugs that have been used to treat malaria and autoimmune diseases; antibodies from people who have recovered from COVID-19, and mechanical respiratory support have been used [5,6,7], specific treatment for COVID-19 in respiratory viral infections pandemics is still needed. According to the Centers for Disease Control and Prevention (CDC), the risk groups include elderly and individuals from all ages who present medical conditions such as diabetes, obesity, chronic kidney and liver disease, hypertension, lung disease, asthma, and any condition that may cause immunosuppression. Since the severity of the infection may be affected by different factors, it is essential to focus on a way of prevention and management of SARS-CoV-2 infection [8]. The identification and protection of susceptible individuals through new molecular targets and pathways for therapeutic intervention is critical. Therefore, in this hypothesis paper, we describe the use of flavonoid supplements to activate the Nrf2 transcription factor, which may be a new target to prevent SARS-CoV-2 infection, reducing oxidative stress and inflammation, enhancing the innate immunity, and downregulating angiotensin-converting enzyme 2 (ACE2) receptors.

## 2. COVID 19 Pathogenesis

The patients infected by SARS-CoV-2 have many symptoms that vary in severity, including dry cough, sore throat, myalgia, fatigue, diarrhea, and shortness of breath [1,9,10]. In patients presenting a severe progression of the disease, acute lung injury (ALI), acute respiratory distress syndrome (ARDS), respiratory failure, heart failure, sepsis, and sudden cardiac arrest were observed within a few days [1,11]. Pathologic assessment of mild COVID-19-contaminated lungs exhibited edema, proteinaceous exudate with globules patchy inflammatory cellular infiltration, and moderate formation of hyaline membranes [12]. In those patients presenting severe cases of ARDS, the postmortem analysis showed bilateral diffuse alveolar damage with edema, pneumocyte desquamation, and hyaline membrane formation [13], resembling the same pathological characteristics seen in SARS- and MERS-induced pneumonia [14].

During SARS-CoV infection, a remarkable inflammatory process is activated by the production of pro-inflammatory cytokines and epithelial and endothelial cell death, leading to vascular leakage, abnormal T cells, and macrophages responses that stimulate ALI/ARDS or even cause death [15]. COVID 19 patients present lymphopenia and pneumonia, and in severe cases, the expression of pro-inflammatory cytokines IL-2, IL-7, IL-10, G-CSF, IP-10, MCP-1, MIP-1A, and TNFα is enhanced [1]. Clinical data indicate that a highly pro-inflammatory state may play a vital role in disease progression and severity. Increased levels of cytokines can induce viral sepsis and lung injury due to inflammatory processes that lead to many complications, such as ARDS, pneumonitis, respiratory failure, shock, organ failure, and ultimately death [16]. A higher level of pro-inflammatory cytokines in the serum of patients with SARS-CoV and MERS-CoV infection was also observed, indicating a similar mechanism of disease severity through cytokine storm [17,18].

Upon viral infection, the body’s innate immune response depends on type I interferon to activate a downstream cascade that can control viral replication and stimulate a successful adaptative immune response. Once in the human respiratory tract, SARS-CoV-2 binds to alveolar epithelial cells and activates the immune system [19]. In healthy people, the viral infection initiates an inflammatory process and recruits immune cells to eliminate the pathogen, and patients can recover. However, in elderly and people underlying medical problems, an uncontrolled immune response can cause an overproduction of cytokines and severe inflammatory disease conditions with increased vascular permeability and a considerable number of blood cells into the alveoli, leading to pneumonia, shortness of breath, inflammation of the airway, dyspnea, and even respiratory failure leading to death [16].

## 3. Nuclear Factor Erythroid-Derived 2-Related Factor 2 (Nrf2)

The transcription factor nuclear factor erythroid-derived 2-related factor 2 (Nrf2, encoded by *NFE2L2* gene) regulates approximately 250 genes involved in cellular homeostasis, including antioxidant proteins, detoxifying enzymes, drug transporters, and numerous cytoprotective proteins [20]. Nrf2 targets genes associated with the cellular defense that contain antioxidant response elements (ARE), which consist of antioxidant enzymes (glutamate-cysteine ligase; GCL), drug-metabolizing enzymes (cytochrome P450s, glutathione S-transferases; GSTs), molecular chaperones, DNA repair enzymes, and proteasome subunits [21]. The transcription of Nrf2-mediated genes depends on Nrf2 heterodimerization with small Maf proteins (MafG, MafK, MafF), which is required for efficient binding to the ARE/EpRE (antioxidant response element/electrophilic response element) [22]. The transcription of these protective genes allows the cell to sustain the redox balance and to eliminate proteins damaged during conditions of oxidative and xenobiotic stress [23]. 

In normal physiological conditions, Nrf2 is found in the cytoplasm associated with the negative regulatory protein, Kelch-like ECH-associated protein 1 (Keap1), which interacts with Nrf2 and works as an adaptor protein, maintaining Nrf2 in low levels [24,25,26], and allowing its continuous degradation through the proteasome in a process mediated by ubiquitin [27,28]. During viral infection, Keap1 detects oxidative stress via the conjugation of redox-sensitive cysteine residues (Cys151, Cys273, Cys288), and Nrf2 is released from Keap1-mediated repression [29,30]. The dissociation from Keap1 prevents Nrf2 ubiquitination, increasing its half-life from 15 to 180 min [31,32,33]. After the translocation of Nrf2 to the nucleus, it creates a complex with coactivators and binds to the promoter region (AREs). This binding induces the transcription of cytoprotective and detoxifying genes [34,35,36,37]. Moreover, Nrf2 activation enhances the innate immune system activity in attenuating or eliminating numerous bacterial and viral pathogens [38] (Figure 1). 

Many studies have described that the excessive damage of cellular macromolecules by numerous factors may induce a feed-forward regulatory loop that leads to cell death [39,40]. The response of the Nrf2-mediated antioxidant system is related to its intracellular regulation, and at low doses of Nrf2 activators, most effectors of the Nrf2/ARE pathway provide cytoprotection [41]. Zucker et al. (2014) suggested a regulatory mechanism of ROS via the Nrf2-dependent feed-forward model. In conditions showing a low oxidative stress level, Nrf2-dependent antioxidant genes are activated, reducing ROS to not harmful levels to cell survival. However, under conditions with an excessive level of ROS, the continuous activation of Nrf2 causes its accumulation in the nucleus and subsequent binding to the Kruppel-like factor 9 (Klf9) promoter. The upregulation of Klf9 transcription, which is a novel regulator of intracellular ROS, could then lead to increased levels of ROS and subsequent cell death [41,42]. Therefore, according to the studies, an excessive amount of oxidative stress is needed to activate the Klf9 transcription, which will be unlikely with a viral infection since the virus needs to keep oxidative stress at an optimal level to maintain viral metabolism without causing host cell death [43].

## 4. Nrf2 Activation and COVID-19 Infection

During the last decades, many studies have been published on Nrf2 role in inflammation, immunity, redox homeostasis, aging, neurodegeneration, and many other areas, but not much has been discussed about its involvement in viral infections. Recently, a study with 40 patients showed the severity of COVID-19 infection directly linked to age and intensity of inflammatory response, inversely associated with Nrf2 expression [44,45]. McCord et al. (2020) proposed that in young healthy individuals, there is a robust oxidative stress-induced activation of Nrf2 that saves host cells from irreversible self-inflicted damage. In elderly or individuals presenting comorbidities involving chronic inflammation, the Nrf2-activation response may be insufficient to break the continuous cycle of events. The induction of Nrf2 activation might allow these individuals to stop cytokine production and begin the recovery and repair stage after the inflammatory process [46]. Pharmacological Nrf2 activation was able to modulate 42 genes associated with respiratory virus infectivity and resistance, or to the associated immune response. Nrf2 activation seems to provide numerous antiviral effects that may grant a degree of resistance, attenuating viral replication rate, ameliorating symptoms, and enabling successful navigation through the cytokine storm, which is a particular problem with COVID-19 [46]. 

Tests performed with lung biopsies from COVID-19 patients showed suppression of the Nrf2 pathway, but on the other hand, Nrf2 pharmacological inducers inhibited the replication of SARS-CoV2 and decreased levels of the inflammatory response [47]. Although redox homeostasis in infected cells and lung inflammation are characteristics of infections caused by respiratory viruses and information obtained from viruses that affect the airways may be pertinent for extrapolation to COVID-19, recent studies with more experimental evidence are emerging [48,49]. Nrf2 activators, such as dimethyl fumarate and 4-octyl itaconate [50], suppressed the inflammatory response to SARS-CoV2 in human cells, including peripheral blood mononuclear cells from COVID-19 patients [47]. Nrf2 presents a high integrated control of the inflammatory response and is necessary for its execution and resolution. Moreover, by controlling the endogenous cytoprotective systems, Nrf2 may play a more physiological function in achieving a balance between beneficial and detrimental effects of inflammation [49].

## 5. Viral Infection and Oxidative Stress

Numerous publications have described the role of viruses in inducing oxidative stress to facilitate their genome replication inside the cell. The infection-induced oxidative stress caused by the virus activates critical antiviral signaling, including toll-like receptor and interferon pathways in the host cell [51]. During infection, the virus needs to express several proteins inside the host cell in order to replicate. These proteins are responsible for inducing oxidative stress, which is not only essential to the viral life cycle and the assembly of a new virion, but also in the viral pathogenesis [52]. However, high levels of oxidative stress can also be a burden to the host cell. Thus, a virus needs to maintain oxidative stress at an optimal level to support its metabolism, but not too high that it would kill the host cell [43].

Problems to maintain an appropriate redox balance by a host cell can contribute to the viral pathogenesis, causing massive induction of oxidative stress-induced cell death [53,54]. This imbalance between ROS production and antioxidant defense system seems to be in a direct association with the disruption of common cellular physiology [52]. Because of this, an increasing number of publications have shown this virus-induced oxidative stress as one of the most critical pathogenic mechanisms for inflammatory response and tissue injury caused by viral infection [55,56].

To neutralize the harmful effects of oxidative stress, mammalian cells have a unique antioxidative defense system that is supposed to be turned off in normal physiological conditions. Conversely, upon encountering oxidative stress, Nrf2, which is an oxidant-sensitive molecule, is activated and transcriptionally stimulates a series of genes responsible for cytoprotection and detoxification. It is one of the best-characterized antioxidative transcription factors with an oxidants/electrophile-sensor function [57].

## 6. Nrf2 Activation Downregulates ACE2 Receptor Expression

Genetically, the sequence of SARS-CoV-2 is ≥ 70% similar to that of SARS-CoV, and both of them have a high affinity for the same receptor located on respiratory epithelial cells, known as angiotensin-converting enzyme 2 (ACE2). ACE2 is an essential receptor for the surface spike (S) glycoprotein [19,58]. The S-protein of coronavirus binds to host receptors and facilitates viral entry into target cells. Through the N-terminal domain and the C-terminal RBD domain, the S-protein directly interacts with the host receptors [59]. In this way, SARS-S binds to the ACE2 receptor and uses the transmembrane protease serine 2 (TMPRSS2) for S protein priming [60]. Although SARS-CoV and SARS-CoV-2 bind to the same ACE2 receptor, the binding affinity of SARS-CoV-2 is about 10 to 20-fold higher than the SARS-CoV, increasing the chances of spread of the disease from one individual to another [61] (Figure 2).

Recent studies reported that Nrf2 activation downregulates ACE2 expression, and its deficiency up-regulates the ACE2 receptor [62]. The investigation used Nrf2 knockout mice and showed that Nrf2-deficient animals presented an enhancement in ACE2 expression. In cultured immortalized renal proximal tubule cells, the transfection with Nrf2 small interfering RNA or treatment with Nrf2 inhibitor (trigonelline) prevented high glucose stimulation of Nrf2 nuclear translocation with an increase in ACE2 transcription [62]. Although the precise mechanism by which a lack of Nrf2 leads to increased ACE2 gene expression is still unclear, the data suggest a potential role of Nrf2 in the modulation of ACE2 receptors and SARS-CoV-2 infection (Figure 2).

## 7. Oxidative Stress-Mediated Hypomethylation Increases Expression of ACE2 Receptor 

ACE2 is a functional receptor that allows SARS-CoV-2 entry into the cells. In SARS-CoV-2 infection, the overexpression of ACE2 receptors may induce viremia and organ damage. During SARS-CoV infection, viral particles, which also bind to the ACE2 receptor, infect immune cells such as the peripheral blood T cells, leading to viral dissemination [63]. ACE2 gene is located on the X chromosome, and the analysis of the whole-genome DNA methylation data showed a substantial hypomethylation in the ACE2 gene in a demethylated T cell subset [64,65]. The hypomethylation implicated CpG sites in the ACE2 promoter region proximal to the transcription start site, the 5′-untranslated region, and the 3′-untranslated region, indicating a functional regulatory effect of the methylation changes [66]. The data showed that ACE2 expression is regulated by DNA methylation and that in lupus disease, a methylation defect may result in ACE2 overexpression [66]. 

Moreover, a more evident DNA methylation defect was observed during increased lupus disease conditions, which could be aggravated by increased oxidative stress levels, such as during viral infections [67]. Oxidative stress was shown to inhibit DNA methylation leading to the attenuation of MEK/ERK signaling and a decrease in the DNA methyltransferase (DNMT1) expression [67,68]. Additionally, oxidative stress lead to mTOR activation and increased DNA methylation defect by inhibiting DNMT1 due to a depletion of NADPH and glutathione levels [69]. Thus, the studies suggest that oxidative stress from SARS-CoV-2 infection may exacerbate DNA methylation defect in lupus patients and further boost viral entry through increased ACE2 expression [66].

During SARS-CoV-2 infection, the response to viral entry in the host may result in tissue damage and oxidative stress, worsening the inflammatory process and leading to cytokine storm [66]. In this way, SARS-CoV-2 infection severity may be aggravated by epigenetic dysregulation, and the dissemination of the disease may be increased because of the overexpression of ACE2 receptors in peripheral blood mononuclear cells [66]. The study indicated that patients with lupus disease are more susceptible to SARS-CoV-2 infections and its complications [66]. Therefore, the investigations indicate that oxidative stress plays a role in the expression of ACE2 receptors and may exacerbate SARS-CoV-2 infection severity.

## 8. Nrf2 Downregulation, Oxidative Stress, and Proteases Expression

Numerous human viruses, including SARS-CoV and influenza, present regulated proteolysis necessary for viral spread/propagation. In the lung, the expression of proteases, TMPRSS2, human airway trypsin-like protease (HAT), and secretory leukocyte proteinase inhibitor (SLPI), are necessary to cleave the viral hemagglutinin surface protein and allow viral fusion and entry into the host cell [70,71,72,73,74]. Interestingly, the SARS-CoV-2 genome lacks the hemagglutinin-esterase gene, which is characteristically found in beta coronaviruses [75]. Studies have demonstrated that an association between oxidative stress and inflammation processes may cause alterations in the expression of these proteases/antiproteases [76,77]. In influenza infection, the increased viral entry and following replication is associated with a decrease in Nrf2 expression/ activity and are mediated by the activation of transmembrane proteases in the cellular host [78]. In asthmatics, HAT is released into the airway fluids during inflammation [76,77], and the gene expression of SLPI was shown to increase in Nrf2-deficient mice, increasing inflammation and showing a balance between oxidative stress and protease expression [79]. Kesic et al., 2011, demonstrated that the decrease in *Nrf2* gene expression induces oxidative stress and stimulates serine protease activity, leading to hemagglutinin cleavage and hence an increased virion entry [78]. The data showed that the activation of *Nrf2* gene expression potentially protects cells from a viral infection, stopping viral entry and replication. The increased expression of Nrf2 may reduce oxidative stress and consequently decrease serine protease activity. Therefore, according to the studies, the transcription factor Nrf2 may play a key role in dictating susceptibility to viral infection at the level of the epithelium [78].

## 9. Nrf2 Activation Modulates Viral Infection Severity

Mechanisms of viral infection are very complex, and studies of factors that can influence the vulnerability to viral respiratory infection are still necessary. Pre-existing diseases, impaired immune response, obesity, and age, among other factors, may affect the severity of the disease [22,80,81]. The replication of the viral genome and new virion assembly is supported by the expression of many proteins by the virus inside the host cell, and many of them are responsible for ROS generation [82,83]. Virus-induced oxidative stress, ROS generation, and pathogenesis of the disease play a vital role in the viral life cycle, increasing the activation of antioxidative defense mechanisms by the host cell. Oxidative stress is caused by an imbalance between ROS production and the body’s ability to detoxify the reactive intermediates readily. In this regard, Nrf2 plays a crucial role in cytoprotection and detoxification in response to oxidative stress, regulating enzymes such as heme oxygenase 1 (HO-1), which belong to phase II antioxidants [43]. The stimulation of antioxidant enzymes is capable of preventing or slowing down oxidative damage to cells. Recent studies indicate Nrf2 as having a pivotal role in cancer progress, chronic lung disease, protection against oxidants, and host defense against viral infections that affect the respiratory tract [84,85,86].

Epithelial cells are the primary target and the main font of antioxidant enzymes during respiratory viral infection and replication. They produce several cytokines, such as type I interferons (IFN-α and IFN-β) and other components of the host innate immune response, in response to infection [87,88]. A double-stranded RNA recognition, produced by viruses during their replication process, is the primary signal for type I IFN synthesis. IFN-α/β, released by the infected cells stimulate the action of mediators involved in the shutoff of viral replication inside the host cell, but also protect neighboring cells that have not been infected yet. An additional characteristic of the respiratory viral infection is its sensitivity to IFN-inducible Mx GTPases (MxA) inhibitory effects, which significantly abolishes viral replication [87,89].

The Nrf2-dependent antioxidant role in the modulation of the interferon/antiviral response in epithelial cells has not been investigated yet. However, some studies show that Nrf2-specific shRNA efficiently reduced both *Nrf2* mRNA and protein expression in epithelial cells, showing a significant increase in viral entry and replication in human nasal epithelial cells. Furthermore, an inverse relationship between levels of Nrf2 expression and susceptibility to viral infection was observed. The results demonstrated the protective role of *Nrf2* gene expression against influenza virus at the level of the epithelium [78]. Genetic and pharmacological manipulation of *Nrf2* expression was shown to modulate influenza virus entry. The suppression of *Nrf2* expression enhanced the entry of the influenza virus, and the increased *Nrf2* expression decreased viral entry, therefore protecting from viral infection [78].

## 10. Nrf2 Activation Enhances Innate Immune Response

The innate immune system provides mechanisms to mediate and prevent infections through the recognition of specific pathogens, such as viruses, bacteria, and fungi [90,91]. It consists of an array of diverse cell types, including monocytes, neutrophils, eosinophils, macrophages, mast, and natural killer (NK) cells, which respond to the pathogen or tissue damage [91]. In innate immunity, the first line of defense is provided by epithelial cells, and in the respiratory system, the various bronchial epithelial cells have a crucial role in primary antimicrobial defense [92].

Transcription factors, such as Nrf2, have been involved in the regulation of defense mechanisms. Nrf2 stimulated innate immune responses that can suppress or eliminate several bacterial and viral pathogens [93,94]. Nrf2 gene-specific knockdown studies using Nrf2 null and wild type (WT) mice as a model to *Streptococcus pneumoniae*-induced pneumonia, identified 53 Nrf2 specific genes and pathways after *S. pneumoniae* instillation in the lung of WT mice [95], but not in the Nrf2 null mice. Nrf2 enhanced innate host defense in Nrf2 null mice compared to WT controls, with raised levels of phagocytosis, Fc receptor effector function, complement activation, and immune globulin regulation [95].

During influenza virus infection, epithelial cells are targeted in the upper respiratory tract, and the defense response is mediated by innate and adaptive immune cells. In response to infection, NK cells kill infected epithelial cells and produce the antiviral cytokine IFN-γ [96]. Although NK cells express or up-regulate the expression of activating receptors to build up antiviral responses, the cells infected by the virus can evade NK cell-mediated recognition. Moreover, the virus can downregulate NK cell-activating receptor ligands and increase engaging inhibitory receptors [97].

Nrf2 activation participates in the priming of NK cells that mediate innate defense in response to viral infection. A recent study investigating the myeloid cell-derived IL-27 signaling reported that IL-27 mediates the regulation of NK cells effector responses. Subsequent to influenza infection, WT mice presented CD27^+^CD11b^+^ effector NK cells in the alveolar space and lung tissue, which was not observed in *IL27ra*^−/−^ mice, suggesting a critical role for IL-27 in regulating this NK subset [98]. It has also been reported that in NK cells, the expression of CD27 may determine the ability of cell migration, suggesting that IL-27 may promote the trafficking of a subset of NK cells to the infection site [99,100]. NK cells from *IL27ra*^−/−^ mice animals showed reduced levels of MafF expression, which is one of the Nrf2 interaction sites for the transcription of cytoprotective genes [98]. Therefore IL-27 seems to have a critical function in NK cell-mediated functions through transcriptional pathways regulated by Mafs and Nrf2 and may play a role in the regulation of adaptive immune response that can determine the pathophysiological outcome after infection by influenza [98].

## 11. Nrf2 Reduces Oxidative Stress and Inflammation 

Upon oxidative stress or in the presence of electrophilic compounds, cells must rapidly increase their antioxidant capacity to maintain homeostasis and try to counteract the enhanced production of ROS. The activation of Nrf2 signaling enhances the expression of Nrf2-target genes that encode crucial protective enzymes: NAD(P)H: quinone oxidoreductase 1 (NQO1), HO-1, GCL, GST, catalase (CAT), superoxide dismutase (SOD) aldo-ketoreductase, γ-glutamyl cysteine ligase, thioredoxin, and thioredoxin reductase [101,102,103,104,105].

Nrf2 controls the redox homeostatic gene regulatory network, and its activation increases the expression of several cytoprotective enzymes that restore redox homeostasis. Nrf2 is mostly linked to antioxidant and detoxification enzymes, cellular transporters [106,107], enzymes that exclude the entry of xenobiotic metabolites and toxic compounds [106], and several components of the proteasome [108]. Nrf2 induces the expression of genes implicated in cell growth, cell adhesion, protein folding, cell signaling, cell-cycle control, survival, and glucose metabolism [108,109]. Nrf2 also promotes the expression of molecular chaperones/heat shock proteins and wound healing response proteins [110].

Nrf2 not only regulates mechanisms of cell defense against oxidative stress, but its activation also induces anti-inflammatory effects and plays a critical role in the resolution of inflammation [50,94,111]. Together, Nrf2 and NFƙB pathways control stress and inflammatory responses. Studies indicate that Nrf2 may counteract NFƙB-driven inflammatory response in numerous experimental models [112,113,114]. Upon activation by lipopolysaccharides, NFƙB is stimulated to induce Nrf2, which up-regulates the expression of HO-1, leading to a decrease in NFƙB inflammatory activity [115,116]. In in vivo studies using Nrf2–/– mice infected with a respiratory syncytial virus, the animals presented elevated virus-induced inflammation, increased mucus cell metaplastic changes, and had a reduction in viral clearance; whereas pre-treatment of Nrf2–/+ or Nrf2–/– mice with an effective Nrf2-activator lead to elevated levels of antioxidants, enhanced viral clearance, and diminished virus-associated inflammation [54].

During SARS-CoV-2 infection, the deterioration of some patients has been related to the so-called “cytokine storm,” which is a type of systemic inflammation induced by infection when white blood cells become activated and produce inflammatory cytokines [117]. While inflammation is crucial for an effective immune response, in SARS-CoV-2, an excessive and prolonged cytokine response may cause ARDS or multiple-organ dysfunction, leading to physiological worsening and death [118]. Clinical studies have detected an increased level of cytokine expression in critical patients with COVID-19, and thus, cytokine’s suppressors may prevent the deterioration of patients, reducing the mortality rates [119].

Investigations of the molecular basis of the Nrf2 function showed that Nrf2 functions not only as a critical regulator of intracellular ROS but also in the regulation of pro-inflammatory cytokine expression [111]. Nrf2 inhibited LPS-induced *IL6* and *IL1b* gene expression through the ROS-independent transcriptional inhibition, and its binding to the proximities of *IL6* and *IL1b* genes implied that Nrf2 inhibited transcription through direct DNA binding. Moreover, the studies indicated that Nrf2-mediated inhibition of the inflammatory cytokine gene expression in M1 macrophages is ARE-independent [111]. Although the precise mechanism of how Nrf2 downregulates target genes that encode inflammatory cytokines is not entirely elucidated, it is clear that increased levels of Nrf2 might help to break the self-perpetuating cycle of events that lead to cytokine storm and help COVID-19 patients to recover faster (Figure 3).

## 12. Nrf2 Low Expression Levels in Elderly

Aging has been identified as a chronic-low grade inflammatory stage, which predicts vulnerability to pathologies related to age [120,121]. A decline in Nrf2 protein and mRNA expression in various tissues, including brain and heart, is observed as we age. This is related to an increase in NFƙB target genes (*ICAM-1* (intercellular adhesion molecule 1) and *IL-6* (interleukin-6)) along with a decrease in Nrf2 target genes (*NQO1*, *γ-GCS*, *HO-1*) [122,123].

Lower nuclear content and decreased Nrf2 activation were observed in senile animals and older adults (>65 years) compared to younger individuals [124,125]. The time that Nrf2 remained active depended on the stimulus created by the inductor, and also on the age of the animals [126]. Several in vivo models, using Nrf2 knockout mice, have presented an increased inflammatory response. In contrast, Nrf2 pathway activation reinstated the redox balance inducing cell repair and limited the generation of free radicals and tumor necrosis factor [79,127,128]. During cellular senescence, there is a decline in the Nrf2 function, and its silencing caused early senescence in human embryonic fibroblasts [129] with a short life expectancy [130]. Nrf2 regulates numerous enzymes, such as superoxide dismutase 1 (SOD1), which may prevent senescence and inflammation in some conditions [131,132]. Nrf2 activation not only induces antioxidant response but also participates in inflammation and might have an essential function in cellular protection and homeostasis [22].

In addition, a reduction in Nrf2 expression seems to mediate a noticeable reduction in neural stem/progenitor cell survival and regeneration during middle age, suggesting that aging may be associated with a decline in Nrf2 expression and the Nrf2 compensatory response to oxidative stress [133]. Therefore, the reduction in Nrf2 levels may be related to the higher severity of SARS-CoV-2 infection disease in elderly people.

## 13. Nrf2, Inflammation, and Obesity

According to the CDC, obesity is among the risk factors that can augment COVID-19 severity. Furukawa et al. (2017) described oxidative stress as one of the main factors involved in obesity-related morbidity, and Nrf2 seems to be a promising new target to treat obesity [134]. Nfr2 seems to have an essential role in inducing preadipocytes to adipocytes differentiation [135,136,137]. The activation of Nrf2 reduces adipogenesis or inhibits total triglycerides content in mature differentiated adipocytes [136]. Nrf2 function in the metabolism of lipids and glucose occurs via regulation of the expression of numerous adipogenic and lipogenic genes, including *FABP4*, *CEBPA*, *CEBPB*, *SREBF1*, *PPARG*, *fatty acid synthase* (*FASN*) and acetyl-CoA carboxylase (*ACACA* and *ACACB*), altered function of the pentose phosphate pathway, and NADPH production. Lack of Nrf2 impairs the activation of Akt protein and the glucose transporter type 4 (Glut4) activity in response to insulin [138]. In vitro studies using adipocytes indicate that the activation of Nrf2 in obesity may be beneficial and ameliorates obesity-induced inflammation in both fat cells and co-cultured macrophage [81]. Nrf2 activation modifies the function of adipocytes and prevents metabolic dysregulation and insulin resistance in lipodystrophic mice via repression of hepatic enzymes for de novo lipogenesis. At the same time, in Nrf2-deficient animals, this effect was not observed [135]. Additionally, activation of ERK/JNK signaling pathway and p38 mitogen-activated protein kinase, as well as AhR inhibition and elevated expression of estrogen receptors, are associated with Nrf2 mediated anti-inflammatory systemic effects in obesity [81]. 

Studies using a mouse model in a high-fat diet (HFD) showed that changes in levels of oxidative stress, impaired glucose disposal, insulin signaling, and obesity development could be reversed by oltipraz, an Nrf2 activator [139]. A long-term HFD feeding led to a diminished nuclear content of Nrf2 in adipose tissue with decreased Nrf2 target protein levels, suggesting that HFD can lead to a defect in the endogenous Nrf2 antioxidant system, which plays a role in the impairment of insulin signaling and energy homeostasis [139]. Animals that received Oltipraz administration became resistant to HFD-induced obesity. Since Nrf2 activation inhibits adipocyte differentiation, the benefits of an Nrf2 activator administration on HFD-induced obesity seems to be through the direct inhibition of adipocyte differentiation [140,141]. The studies indicated an essential role of the endogenous Nrf2 antioxidant system in the prevention and development of insulin resistance and obesity. They suggested that the induction of Nrf2-dependent antioxidant enzymes may decrease oxidative stress and be a potential approach to fight obesity and the insulin resistance associated with it [139].

## 14. Nrf2 Reduces Oxidative Stress Associated with Hyperglycemia

Recent studies described obesity-related complications, such as diabetes, as a risk factor to COVID-19 severity. According to recent reports and the CDC, there is a link between severe obesity and coronavirus mortality. Although studies published still have a small number of subjects, Stefan et al. (2020) described that 85% of the patients with obesity required mechanical ventilation and 62% of the patients with obesity died, compared to 64% that required mechanical ventilation and 36% that died among those patients without obesity. Considering that diabetes mellitus (DM) is highly associated with increased adipose tissue mass, the study demonstrated that a high BMI might be a critical risk factor for a severe state of disease in these patients [142].

Diabetes mellitus is comprised of many metabolic disorders due to mechanisms of failure in insulin secretion or activity, leading to hyperglycemia and other complications. DM type I is caused by an autoimmune destruction of pancreatic β-cells, resulting in a lack of insulin and diabetes [143]. DM type 2, which is non-insulin-dependent diabetes, is presented in individuals genetically predisposed and increases with age. It is linked to resistance to insulin and hyperinsulinemia, β-cell failure, and subsequent deficiency of insulin [144]. DM type 2 is characterized by hyperglycemia, hyperinsulinemia, and inflammation, which lead to a pro-oxidative milieu with increased production of ROS and a decreased expression of antioxidant enzymes [145,146,147,148,149,150,151]. Thus, the upregulation of genes that code for detoxification, antioxidant, and anti-inflammatory mediators may be a potential therapeutic strategy to protect against inflammation and oxidative stress that are enhanced in DM [152].

Nrf2 activation can increase the expression of what may counteract the pro-oxidative condition by directly detoxifying ROS, elevating the cellular antioxidant defense, improving mitochondrial function, inducing the expression of enzymes of the pentose phosphate pathway, preserving endothelial function, and decreasing the levels of blood glucose [153,154,155,156]. Studies showed that Nrf2 activation was higher in the early phases of DM and lowered at later stages, and those Nrf2 deficient animals showed deterioration in diabetic symptoms and further complications [157]. Moreover, in the context of hepatic insulin resistance, bardoxolone-methyl (CDDO-Me), a derivative of the natural oleanolic acid, showed to be promising in clinical trials with patients suffering from diabetic chronic kidney disease through Nrf2 activation [158]. With Nrf2 being a major transcription factor in the production of cytoprotective and antioxidant enzymes, it seems to be a potential target to attenuate oxidative stress associated with hyperglycemia [159].

## 15. Flavonoids Role as Antivirals and Nrf2 Activators

Flavonoids are a family of polyphenolic compounds found in plants that are responsible for a range of pharmacological properties, including antioxidant, anti-inflammatory, anticancer, antibacterial, antifungal, and antiviral activity [160]. Flavonoids antiviral activities were reported against several viruses, including HSV-1, HSV-2, human cytomegalovirus, and some types of human adenoviruses [161,162,163]. Many of these polyphenolic compounds, including apigenin, luteolin, quercetin, amentoflavone, daidzein, puerarin, epigallocatechin, epigallocatechin gallate, and gallocatechin gallate, presented antiviral activity through inhibition of the proteolytic activity of SARS-CoV 3C-like protease, which is vital to viral replication [164,165].

Although many flavonoids may activate the transcription factor Nrf2, studies using two flavonoids, in particular, have demonstrated the great potential of these two compounds, not only as Nrf2 inducers but also because of their strong anti-viral activity. They are epigallocatechin-3-gallate and thymoquinone, which are already in the market and sold as daily supplements.

### 15.1. Epigallocatechin-3-Gallate

Supplementation with flavonoids has been shown to induce Nrf2 gene expression [166,167]. Green tea has been one of the most consumed health-promoting beverages in several countries [168]. Its primary constituent polyphenols (also called catechins) have been described for their antitumor, antioxidative, and antimicrobial activities [169,170]. Epigallocatechin-3-gallate (EGCG) is considered the main active constituent of green tea and accounts for approximately 59% of the total polyphenols in dry green tea leaves. Xu et al. (2017) listed several publications regarding the anti-viral properties of green tea catechins, including the inhibitory effect on DNA virus (HBV, Herpes virus, EBV, and Adenovirus) and RNA virus (HIV, HCV, Influenza virus, some arboviruses, Human T-cell Lymphotropic Virus-1, Rotaviruses and Enteroviruses, EBOV) [171]. EGCG is a potent antioxidant through scavenging ROS and reactive nitrogen species, production of defense enzymes, and chelating and binding of divalent metals [172].

Lambert et al. (2010) evaluated the hepatotoxicity of a high oral dose of EGCG in mice [173]. The results demonstrated that EGCG doses that caused toxicity corresponded to approximately 10.5–32 cups of green tea, which is an excessive daily amount. Therefore, the data did not indicate that tea consumption could present a significant risk for hepatotoxicity. The authors stated that a toxicity risk might exist with high doses of dietary supplements containing concentrated or purified tea preparations [173]. However, these doses themselves are not within the apparent toxic range determined in the study, but they are similar to doses of green tea supplements associated with human hepatotoxicity observed in case-reports and the possibility of reaching the toxic threshold by exceeding the recommended dose is not insignificant [174,175]. 

In a clinical trial, the safety of green tea catechins administration containing 400 mg of EGCG given in divided doses (twice a day) and non-fasted condition was evaluated. EGCG did not cause liver or other toxicities in men at elevated risk for prostate cancer, which were the subjects of the study. The results provided strong evidence that a daily intake of a standardized, decaffeinated catechin mixture containing 400 mg EGCG per day for one year, administered with food (non-fasting), in two divided doses, accumulated in plasma, was well tolerated and did not cause any treatment-related adverse effects [176].

Studies demonstrated that in the absence of infection, EGCG supplementation enhanced the expression levels of Nrf2-dependent genes and antiviral mediators, and blocked virus entry in nasal epithelial cells [78]. Through genetic and pharmacological manipulation of Nrf2 expression in human nasal epithelial cells expressing shRNA that targeted Nrf2, the study demonstrated that suppression of Nrf2 expression augmented virus entry, whereas an increase in Nrf2 expression decreased influenza A virus entry [78].

In the absence of infection, the supplementation with EGCG increased Nrf2 protein levels and induced the mRNA expression levels of antiviral response genes, including *RIG-I*, *IFN-β*, and *MxA* [78]. The expression of these genes is usually stimulated during viral infections where increased production of type I IFNs leads to the synthesis of antiviral genes inducing an “antiviral state” and limiting viral replication [177]. MxA, which belongs to a small family of GTPases, has been described to inhibit viral replication and have antiviral activity toward influenza A virus [177]. RIG-I, a cytosolic DExD/H box-containing RNA helicase (that works together with dsRNA), showed to enhance the production of interferon in response to viral infection [178,179]. The authors hypothesized that EGCG up-regulates the expression of these antiviral genes, which proactively protect the cells before viral infection by creating an “antiviral protective state” [78]. Although binding sites for Nrf2 have not been identified in the promoters of *IFN-β*, *RIG-I*, and *MxA* genes, the study suggests this possibility, and thus the induction of Nrf2 would increase the transcription of these antiviral genes [78].

EGCG has potent antioxidant capacities, stimulating the expression of several antioxidant enzymes [180]. In vitro and in vivo studies demonstrated that EGCG stimulates the expression of phase II antioxidant genes, which are associated with Nrf2-EpRE signaling [166,181,182,183]. Even though Wu et al. (2006) described that the mechanism of activation for Nrf2 induction involves serine/threonine phosphorylation and increased nuclear accumulation and binding to the EpRE [184], it is still not known how the activation of *Nrf2*-dependent gene expression is involved in its potential antiviral activities [78]. The data indicated that the transcription factor Nrf2 is an essential key in dictating susceptibility to viral infection at the epithelium level and that EGCG nutritional supplementation increases Nrf2 protein levels and augments the expression of antiviral mediators in the absence of viral infection [78].

### 15.2. Thymoquinone

The polyphenol thymoquinone (2-Isopropyl-5-methylbenzo-1,4-quinone; TQ) is an import constituent of black cumin (*Nigella sativa*), which has been used for thousands of years as a spice and food preservative, as well as a protective and curative remedy for numerous disorders, and it is known to have many medicinal properties in traditional medicine [185,186]. In a recent review, Molla et al. (2019) described the anti-viral activity of *Nigella sativa* and its constituents against several human, animals, birds and plant pathogenic viruses, such as murine cytomegalovirus infection, avian influenza (H9N2), *Schistosoma mansoni* infection, PPR virus, broad bean mosaic virus, HIV, hepatitis C virus, zucchini yellow mosaic virus, and papaya ring spot virus, suggesting that *Nigella sativa* may be one of the best sources of anti-viral drugs [187]. Furthermore, *N. Sativa* has been established as a safe herbal product [188]. Clinical trials using black seed and its active constituent, TQ, revealed that their administration did not cause liver, kidney, or gastrointestinal side effects [189,190]. Their use showed to be safe in patients with type 2 DM, causing no renal or hepatic problems [191]. Moreover, the administration of *N. Sativa* seeds did not affect the serum alanine aminotransferase (ALT) or the serum creatinine (Cr) levels in adults [192].

TQ has been reported as a potent compound against oxidative stress and inflammation [193] and has been described to induce the expression of many cytoprotective enzymes, such as glutathione S-transferase [194,195,196], glutathione peroxidase [195], glutathione reductase [195], and superoxide dismutase [196]. During disease state involving oxidative stress, cells activate a range of cytoprotective enzymes involved in cellular antioxidant defense, such as HO-1, which is activated and has been shown to improve oxidative and inflammatory tissue damage [197]. In keratinocyte cells, TQ increased HO-1 expression at both mRNA and protein level, and its effect on the Nrf2-mediated signaling pathway demonstrated that TQ-induced HO-1 expression is dependent on the activation of Nrf2. The data indicated that TQ works as a pro-oxidant that leads to the activation of Nrf2 signaling and stimulation of HO-1 expression in HaCaT keratinocyte cells [198]. The mechanism by which TQ stimulates Nrf2 activation and HO-1 expression did not involve MAP kinases, but consisted of the phosphorylation of AMPKα and kinase Akt, and decreased PTEN expression [198], which is a negative regulator of Akt phosphorylation [199].

In BV-2 microglial cells from rats exposed to LPS, TQ inhibited the release of TNF-α, IL-6, and IL-1β, and decreased the levels of TNF-α, IL-6, IL-1β, and prostaglandin E2 (PGE2) [200]. TQ decreased iNOS protein levels, ƙB inhibitor phosphorylation, and binding of NFƙB to the DNA, suggesting that TQ exerts its effect via inhibition of NFƙB-dependent neuroinflammation, which involves NFƙB-mediated pro-inflammatory mediators that participate in inflammation and ROS production [201]. Besides, TQ increased nuclear accumulation of Nrf2, enhanced the binding of Nrf2 to ARE, and increased its transcriptional activity, as well as augmented protein levels of NQO1 and HO-1. The results suggested an association between the TQ activation effect over the Nrf2/ARE signaling pathway and TQ inhibitory effect in NFƙB-mediated neuroinflammation [200].

Studies with SARS-CoV-1 patients and in vivo experiments demonstrated that a CoV-neuroinvasive potential and spread from the respiratory tract to the Central Nervous System (CNS) could happen through retrograde axonal transport from peripheral nerves or hematogenous spread [202]. Data indicated that once in the CNS, CoV could induce neuronal cell death in mice [203]. Li et al. (2020) speculated that SARS-CoV-2 neuroinvasive potential, mainly of medullary structures involved in respiration, may somewhat mediate the elevated incidence of respiratory failure observed in COVID-19 [204]. During SARS-CoV infection, the ACE2 receptor is also expressed in neurons and glia. Experimental studies using intranasally-inoculated SARS-CoV-1 infection in ACE2 transgenic mice showed neuronal death and increased regulation of pro-inflammatory cytokines production by neurons and astrocytes [203]. Peripheral myeloid cells infected by CoV [202] may be recruited or transmigrate to the CNS due to increased blood-brain barrier permeability caused by inflammation or psychological stress. In the CNS, monocytes infected by the virus can disseminate neuroinflammation by releasing inflammatory cytokines and causing microglial activation [205,206]. Evidence suggests that CoV can persistently infect leukocytes [207,208], and thus, the time-course over which CoV-infected immune cells could serve as a prospective font of neuroinflammation could be considerably longer than the early infection and acute symptom state [209].

Considering the harmful effects of oxidative stress and the neuroinvasive potential of coronaviruses infection, the studies described here suggest that TQ treatment may have a potential role by activating Nrf2 and inducing HO-1 expression, and also inhibiting pro-inflammatory cytokines release, which could help in the prevention of COVID-19 infection or reduction of the disease severity.

## 16. Vitamin D3 Supplementation to Enhance Immune Response

Vitamin D has been identified as a nutrient that contributes to the immune system’s health and enhances defense against infections. In the lung, the enzyme 1α-hydroxylase converts vitamin D to its active form 1,25-dihydroxyvitamin D_3_, which is the most potent metabolite of vitamin D [210]. It plays a crucial role in mediating inflammation and immune response. Vitamin D mediates NFƙB signaling and cytokine production during infection controlling airway epithelial cell immune responses [211]. It induces IƙBα, leading to a lower stimulation of NFƙB-dependent genes through viral infection, decreasing the release of inflammatory chemokines [212]. Additionally, vitamin D increases CD14 and cathelicidin expression, which help in the recognition and elimination of viruses [212,213]. A decline in vitamin D levels may lead to a pro-inflammatory phenotype, which may augment disease severity. Lower levels have been linked to a higher vulnerability to infections and associated with increased risks of respiratory diseases, such as asthma, chronic obstructive pulmonary disease, and decreased lung function, affecting the body’s ability to fight respiratory infection [211,214].

Studies demonstrated that individuals with levels of 25(OH)D lower than 16 ng/mL presented more respiratory infections than those with higher levels [215], and those individuals with 25(OH)D level higher than 38 ng/mL presented a considerable decrease in the development of acute respiratory tract infections [216]. Besides, a study with individuals admitted in intensive care units showed that vitamin D deficiency was associated with infection severity, longer treatment length, and increased mortality [217,218]. Furthermore, higher mortality was observed among individuals with acquired pneumonia and who presented 25(OH)D levels lower than 12 ng/mL [219]. The studies show that vitamin D may potentiate the immune response and mediate inflammatory cascades, reducing the chances of infection severity [211]. Additionally, a recent study showed that Nrf2 expression levels were notably increased in 1,25(OH)_2_D_3_-treated mouse embryonic fibroblasts from WT mice, but not in the vitamin D receptor (VDR) knockout mice, and that VDR presented the ability to bind Nrf2. Furthermore, Nrf2 knockdown reduced Nrf2 target genes expression indicating that 1,25(OH)_2_D_3_ have an antioxidant function in Nrf2 transcriptional regulation mediated via the VDR. The data indicate that 1,25(OH)_2_D_3_ deficiency increases oxidative stress, inhibiting transcription of Nrf2, and enhancing DNA damage [220]. Taken together, the studies show that vitamin D supplementation may decrease the risk of severe infections, in particular, the ones that affect the viral respiratory tract, such as SARS-CoV-2.

Recent studies indicate that deficiency of vitamin D may be a much bigger matter than expected, even in sunny locations. Therefore, it may no longer be suitable to assume that the linear latitude gradient is the most significant determinant of vitamin D levels [221]. In the past, it was believed that vitamin D deficiency was a problem restricted to countries located in higher latitudes. However, studies have shown that vitamin D deficiency is a common phenomenon, despite the abundance of sunlight in countries such as Brazil, making it a global health problem [222,223]. 

Therefore, we believe that the ingestion of EGCG and TQ should be combined with vitamin D3 supplementation, even in countries where individuals may have UVB radiation throughout the year, due to its essential role in immune mechanisms against viral infection.

## 17. Conclusions

In this hypothesis paper, we discussed the protective role of Nrf2 transcription factor and its association with SARS-Cov-2 infection severity. Nrf2 plays a critical role in modulating the susceptibility to viral infection at the level of the epithelium and protects cells from a viral infection, reducing oxidative stress, inflammation, and the expression of ACE2 receptors, which are augmented during SARS-Cov-2 infection. Elderly and individuals with pre-existing medical conditions or immunosuppressed present lower levels of Nrf2, which seems to be associated with a higher risk for developing more severe complications of COVID-19 illness. Nrf2 activation may modulate Nrf2-dependent antiviral mediators before viral infection and limit viral entry and replication. Thus, Nrf2 may be a new target to prevent SARS-CoV-2 infection or even reduce the severity of the disease with reduced oxidative stress, amelioration of inflammation processes (that could lead to cytokine storm), enhanced innate immunity, and downregulation of ACE2 receptors. 

Moreover, we suggest a new approach to target and promote Nrf2 activation by using flavonoid compounds in combination with vitamin D3. EGCG and thymoquinone are natural compounds already being used as supplements. They have been described to activate Nrf2-dependent genes that act in a proactively way, stimulating an antiviral protective state in the host. Besides, vitamin D3 intake may potentiate the effects of EGCG and thymoquinone supplementation. Vitamin D was shown to decrease the development of acute respiratory tract infections considerably, mediating inflammation and immune response through modulation of NFƙB signaling and cytokine production during infection. Vitamin D deficiency also increased oxidative stress, inhibiting transcription of Nrf2, and enhancing DNA damage, showing that vitamin D3 intake may be beneficial to activate the Nrf2 transcription factor.

Therefore, in this hypothesis paper, we suggest that the combination of EGCG, TQ, and vitamin D3 may activate Nrf2-dependent genes and protect the cells against viral infection and could be used in the prevention of viral infections, such as SARS-CoV-2. The critical point of this approach is that these supplements would be effective only if taken in combination, which would target and promote Nrf2 activation. Currently, they are already in the market and taken individually, but our suggestion is that vitamin D3 intake may potentiate the effects of EGCG and thymoquinone supplementation, activating Nrf2-dependent cytoprotective genes that act in a proactively way, stimulating an antiviral protective state in the host. Future studies and clinical investigations are necessary to assure the therapeutic effect of EGCG, TQ, and vitamin D3 combination as Nfr2 activators and delineate their clinical benefits in the prevention and management of SARS-CoV-2 infection (Figure 4).

## Figures and Tables

**Figure 1 antioxidants-09-00659-f001:**
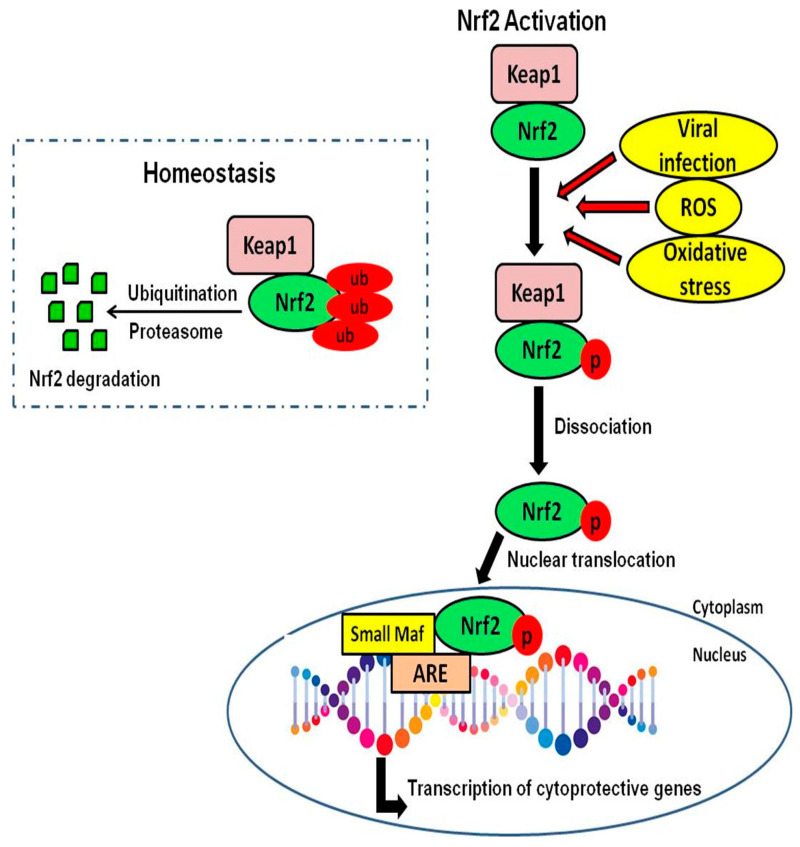
Nrf2 activation. The diagram shows the Nrf2-Keap1 inhibitory complex in homeostasis state and Nrf2 activation under stressful conditions, leading to the transcription of cytoprotective genes.

**Figure 2 antioxidants-09-00659-f002:**
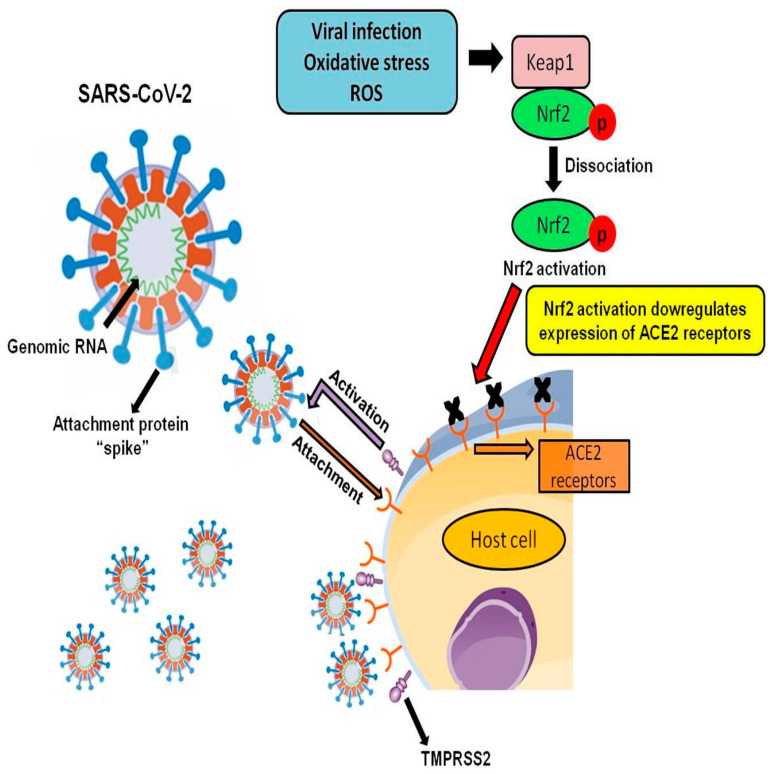
Nrf2 downregulatory effect on ACE2 Receptors during severe acute respiratory syndrome coronavirus 2 (SARS-CoV-2 infection). The diagram illustrates SARS-CoV-2 binding to its ACE2 receptors in the host cell through its attachment protein “spike” and concomitant TMPRSS2 activation, initiating the infection process. The figure also presents Nrf2 activation and subsequent decrease in ACE2 receptor expression, as described by Zhao et al., 2018 [62].

**Figure 3 antioxidants-09-00659-f003:**
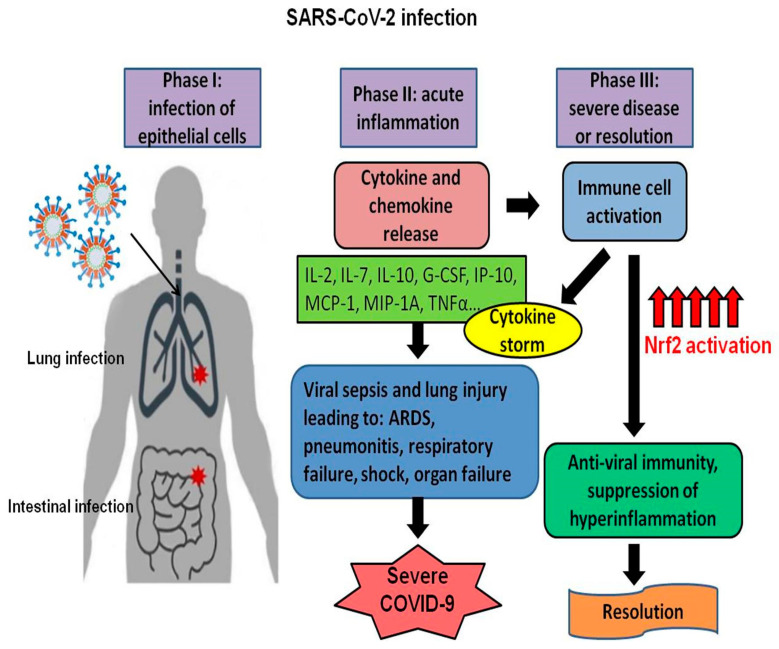
Nrf2 activation limits cytokine production and reduces SARS-CoV-2 infection severity. The diagram shows the phases during SARS-Cov-2 infection and how it can lead to severe disease or patient recovery through the activation of the Nrf2 transcription factor.

**Figure 4 antioxidants-09-00659-f004:**
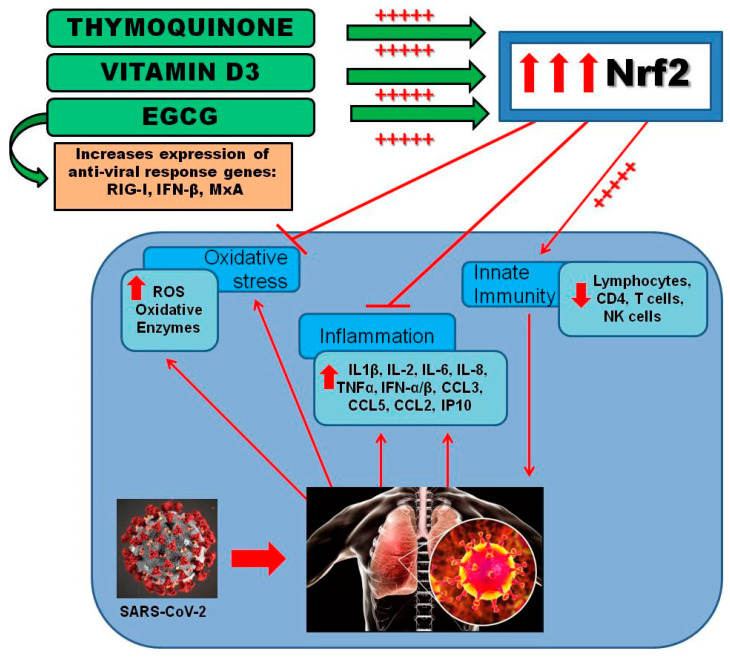
Suggested Nrf2 activation by the combination of EGCG, TQ, and vitamin D3. The diagram shows the activation of Nrf2 by EGCG, TQ, and vitamin D3 given in combination, reducing oxidative stress, inflammation, and stimulating innate immunity as a way to prevent or modulate SARS-CoV-2 infection.

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
