# Peer review of "Flavonoids Activation of the Transcription Factor Nrf2 as a Hypothesis Approach for the Prevention and Modulation of SARS-CoV-2 Infection Severity"

_antioxidants, 2020, doi:10.3390/antiox9080659_

Round 1

Reviewer 1 Report

In this manuscript, Mendonca and Soliman reports the use of flavonoid supplements combined with vitamin D3, which may be a new target for preventing and / or reducing the severity of SARS-CoV-2 infection by reducing oxidative stress and inflammation, increasing congenital immunity and ACE2-lowering receptors. The authors propose a new approach to target and promote Nrf2 activation that modulates virus-induced oxidative stress, ROS generation, and disease pathogenesis.

The topic is relevant and research in the article shows that epigallocatechin-3-gallate, thymoquinone, and vitamin D3 can be used to prevent viral infections such as SARS-CoV-2.
The paper is well written and the text is clear and easy to read. The topic of this manuscript may meet the reader interest of the journal and is especially relevant in the COVID-19 pandemic.

Author Response

No recommendations were provided

Reviewer 2 Report

Dear authors, 

the review "Flavonoids Activation of the Transcription Factor Nrf2 as a Potential Approach for Prevention and Modulation of SARS-CoV Infection Severity" is focused on a high interesting and current topic. 

I suggest to make a figure to describe the putative/described mechanism of action of flavonoids activation in cell.

Moreover, I think could be useful to highlight as in the text as in the conclusion the main critical point of the research about the topic of the review.

Author Response

Reviewer 2:

I suggest making a figure to describe the putative/described mechanism of action of flavonoids activation in the cell. Moreover, I think could be useful to highlight as in the text, as in the conclusion, the main critical point of the research about the topic of the review.

Response: We appreciate and accept the reviewer's suggestion. A new figure describing the main action of flavonoids plus vitamin D3 was added at the end of the review, as a summary of our conclusions.

The critical point of this review is that these supplements (EGCG, thymoquinone, and vitamin D3) must be taken in combination in order to be effective in targeting and promoting Nrf2 activation. Currently, they are already in the market and taken individually, but our approach is that vitamin D3 intake may potentiate the effects of EGCG and thymoquinone supplementation, activating Nrf2-dependent cytoprotective genes that act in a proactively way, stimulating an antiviral protective state in the host.

As suggested, we highlighted this critical point in the conclusion of the review (Conclusion, lines: 657 - 662).

Reviewer 3 Report

The manuscript submitted to Antioxidants by P. Mendonca and K.F.A. Soliman aims to present a review of the main biological involvements of the NFE2L2/ARE pathway and their relationship with SARS-CoV infection. The review of NFE2L2/ARE pathway implication in several biological features is per se interesting and well documented. The second part concerning SARS-CoV infection looks fine but I’m not a specialist of viral infection. However, the link between the two is unclear and is mainly based on the extrapolation of disparate data obtained from various types of viruses and various human or mouse data, cellular or animal models. Thus, rather than a review this manuscript should be classified as a "hypothesis".

In addition, some major specific concerns have to addressed.

1) The nomenclature of gene/protein names must follow the current rules of writing, i.e. italicized upper case letters for human genes and non-italicized upper-case letters for human proteins; lower-case letters will be used for murine or other species gene/protein names.

2) “Nrf2” must be changed to NFE2L2 throughout the manuscript.

3) “klf9” (KLF9?) must be defined and described in order to understand its role and importance.

4) Lines 111-112 “Therefore, according to the studies, an excessive amount of oxidative stress is needed to activate the Klf9 transcription, which will be unlikely with a viral infection.”: what “studies” do the authors refer to? The authors cite only one study [33], which is not related to a viral infection in humans but deals with bleomycin-induced pulmonary fibrosis in mice. In addition, what supports the hypothesis that “will be unlikely with a viral infection”?

5) In the same way, lines 156-157 “Moreover, a more evident DNA methylation defect was observed during increased lupus disease condition, which could be aggravated by increased oxidative stress levels, such as during viral infections [43].” what supports the hypothesis that viral infection increases oxidative stress. Based on several references in the literature, authors should emphasize the link between viral infection and oxidative stress, which is not obvious to non-virologist readers.

6) Many parts of the manuscript are redundant and its organization must be revised. For example, §7 repeats much of what has already been said and could perhaps be put back to the beginning. §9 also contains a lot of repetition and could replace other less useful paragraphs or parts of paragraphs.

7) The title of §5 “Oxidative Stress Increases Expression of ACE2 Receptor” does not reflect the content of the paragraph, 70% (18 lines over 26) of which deals with DNA methylation. And what is the link between epigenetic regulation, oxidative stress and NFE2L2?

8) Line 442 “Studies demonstrated that in the absence of infection”, please indicate that it is an influenza A infection.

Author Response

Reviewer 3:

1) The nomenclature of gene/protein names must follow the current rules of writing, i.e., italicized upper case letters for human genes and non-italicized upper-case letters for human proteins; lower-case letters will be used for murine or other species gene/protein names.

Response: As advised, the manuscript was revised to follow the rules of writing regarding genes and proteins.

2) "Nrf2" must be changed to NFE2L2 throughout the manuscript.

Response: We respectfully disagree that we have to change Nrf2 to NFE2L2 throughout the manuscript. Nrf2 corresponds to a protein, which is a member of a small family of basic leucine zipper (bZIP), and the NFE2L2 corresponds to the gene that codes for Nrf2 protein.

In our review, we are discussing the Nrf2-Keap1-ARE signaling, which refers to the Nrf2 protein in the cytoplasm associated with Keap1 as its negative regulatory protein. We also discuss the disruption of the Nrf2-Keap1 complex during viral infection and oxidative stress, and the translocation of Nrf2 transcription factor to the nucleus, where it binds to ARE domain. So, we believe that the use of Nrf2 is correct. Also, throughout the writing of our review, we kept the same wording found in the original manuscripts.

3) "klf9" (KLF9?) must be defined and described in order to understand its role and importance.

Response: The definition and description of Klf9 were included in lines 118 and 119.

4) Lines 111-112 "Therefore, according to the studies, an excessive amount of oxidative stress is needed to activate the Klf9 transcription, which will be unlikely with a viral infection.": what "studies" do the authors refer to? The authors cite only one study [33], which is not related to a viral infection in humans but deals with bleomycin-induced pulmonary fibrosis in mice. In addition, what supports the hypothesis that "will be unlikely with a viral infection"?

Response: we agree with the reviewer and have included a new reference, which also investigates the role of Nrf2 in Klf9 activation. (References 41 and 42 - line 120).

While these studies are not related to viral infection in humans, they describe the role of Nrf2 in up-regulating the transcription of Klf9, which is a regulator of intracellular ROS. Even though these studies used a mouse and human fibroblasts, and lens epithelial cells, both of them describe the same association between Nrf2 activation and Klf9 expression, which are associated with the modulation of oxidative stress levels.

Chhunchha et al., 2019 (32) demonstrated that toxic doses of sulforaphane (a natural compound) enhanced Nrf2 expression and correlated with aberrant levels of Klf9 expression, causing the acceleration of ROS generation and cell death. However, lower doses of the compound did not affect Klf9 expression and led to stimulation of Nrf2/ARE-mediated cellular protection.

In our review, the purpose of describing the association of Nrf2 and Klf9 was to demonstrate that the activation of Nrf2 is not always beneficial and cytoprotective, and that continuous activation of Nrf2 can aggravate oxidative stress and induce cell death. However, we have to consider that during viral infection, the virus needs to express many of the nonstructural and structural proteins inside the host cell to support its genome replication and the assembly of a new virion. Several of these proteins were described to be responsible for the production of various ROS, and too much oxidative stress will be a burden on the host cell. Therefore, the virus needs to keep oxidative stress in an optimal level to support the viral metabolism without killing off the host cell.

Based on these studies an abnormal/continuous activation and accumulation of Nrf2 would be necessary to cause an aberrant increase in Klf9 expression, which is unlikely with a viral infection since the virus needs to maintain optimal levels of oxidative stress to maintain the host cell alive and take advantage of its machinery (information added to section 3 - lines 121-123).

5) In the same way, lines 156-157 "Moreover, a more evident DNA methylation defect was observed during increased lupus disease condition, which could be aggravated by increased oxidative stress levels, such as during viral infections [43]." what supports the hypothesis that viral infection increases oxidative stress. Based on several references in the literature, the authors should emphasize the link between viral infection and oxidative stress, which is not obvious to non-virologist readers.

Response: We agree with the reviewer that the review was missing more information about the link between viral infection and oxidative stress. A new section was included in the review (please see new section #5).

6) Many parts of the manuscript are redundant, and its organization must be revised. For example, §7 repeats much of what has already been saying and could perhaps be put back to the beginning. §9 also contains a lot of repetition and could replace other less useful paragraphs or parts of paragraphs.

Response: Following the reviewer's suggestion, modifications were made in sections 7 and 9 (currently sections 9 and 11). Some information was removed from these sections and included in section 3 (lines: 101-108).

7) The title of §5 "Oxidative Stress Increases Expression of ACE2 Receptor" does not reflect the content of the paragraph, 70% (18 lines over 26) of which deals with DNA methylation. And what is the link between epigenetic regulation, oxidative stress, and NFE2L2?

Response: the link between epigenetic regulation, oxidative stress, and activation of the Nrf2 transcription factor is a complex mechanism. The purpose of section #5 (current section 7) was to discuss the role of oxidative stress in the expression of ACE2 receptors, which are the ones that allow SARS-CoV-2 entry into the cells and induce viremia and organ damage. Nrf2 activation is linked to this process regarding oxidative stress levels, stimulating several cytoprotection and detoxification genes, and being considered the best-characterized antioxidative transcription factor.

We agree that this section discusses DNA methylation in lupus patients. Still, the goal was to show that SARS-CoV-2 infection severity may also be aggravated by epigenetic dysregulation, which causes oxidative stress and could lead to the worsening of the disease state. A situation that could be regulated by Nrf2 activation. We believe that this is important information indicating that lupus patients are also in the high-risk group, being more susceptible to SARS-CoV-2 infections and its complications.

8) Line 442 "Studies demonstrated that in the absence of infection," please indicate that it is an influenza A infection.

Response: As requested, influenza A was added to line 510.

Reviewer 4 Report

In this review, authors have discussed the protective role of Nrf2 transcription factor and its association with SARS-Cov-2 infection severity. They have shown Nrf2 plays a critical role in modulating the susceptibility to viral infection at the level of the epithelium and protects cells from a viral infection, reducing oxidative stress, inflammation, and the expression of ACE2 receptors, which are augmented during SARS-Cov-2 infection. 

In the review by P. Mendonca, authors described the usage of flavonoid supplements combined with vitamin D3 to activate Nrf2, which they suggested may be used as a new target to prevent SARS-CoV-2 infection and enhance the innate immunity.

  1. The present review is well written and is of interest to readers. COVID-19 pathogenesis and usage of Vit D3 to treat COVID-19 is well written and described but for the rest of the review, authors have discussed modulation of NRF2 activation in other viral infections. Same stands for association of other factors with NRF2 activation.  Thus, I would suggest authors to submit it as mini review or give proper reference or citation to support their conclusions about NRF2 activation in SARS-CoV2 infection.
  2. Please cite the research article with references relevant to COVID-19 for NRF2 activation by contributing factors:
  3. In the sentence “Moreover, we provide a new approach to target ……………………….compounds in combination with vitamin D3” authors must modify the phrase to suggest the approach and tone down the sentence as they are not mentioning about their finding.
  4. Likewise they should modify other sentences with reference to SARS-CoV2 infection.
  5. Few sentences need grammatical correction.

Overall I recommend minor revision and precision in the review.

Author Response

Reviewer 4

  1. The present review is well written and is of interest to readers. COVID-19 pathogenesis and usage of Vit D3 to treat COVID-19 is well written and described, but for the rest of the review, authors have discussed modulation of NRF2 activation in other viral infections. Same stands for association of other factors with NRF2 activation. Thus, I would suggest authors to submit it as mini review or give proper reference or citation to support their conclusions about NRF2 activation in SARS-CoV2 infection.

Response: We agree that the present review discusses the modulation of Nrf2 activation in many other viral infections not caused by SARS-CoV2. However, we also presented data on SARS-CoV and Influenza. Genetically, the sequence of SARS-CoV-2 is ≥ 70% similar to that of SARS-CoV, and both of them have a high affinity for the same receptor, ACE2. Influenza infection and COVID-19 also were shown to share many characteristics. Studies demonstrated that the increased viral entry and following replication Influenza infection is associated with a decrease in Nrf2 expression/ activity and is mediated by the activation of transmembrane proteases in the cellular host.

Because these viruses may have in common the ability to induce inflammation and oxidative stress and also cause overexpression of ACE2 receptors, which are factors that can be regulated by Nrf2 activation, we do believe we can infer that targeting Nrf2 would be a way to prevent SARS-CoV-2 infection or reduce the severity of the disease. Corroborating with our suggestion, Cuadrado et al. (2020) stated that: "Considering that changes in redox homeostasis in infected cells and lung inflammation are hallmarks of infections caused by respiratory viruses (Komaraveli, et al., 2014), the information obtained from viruses that affect the airways may be relevant for extrapolation to COVID-19.

In addition, recently, a study with 40 patients showed the severity of COVID-19 infection directly linked to age and intensity of inflammatory response, inversely associated with Nrf2 expression. Pharmacological Nrf2 activation was able to modulate 42 genes associated with respiratory virus infectivity and resistance, or to the associated immune response. Nrf2 activation seems to provide numerous antiviral effects that may grant a degree of resistance, attenuating viral replication rate, ameliorating symptoms, and enabling successful navigation through the cytokine storm, which is a particular problem with COVID-19. Tests performed with lung biopsies from COVID-19 patients showed suppression of the Nrf2 pathway, but on the other hand, Nrf2 pharmacological inducers inhibited the replication of SARS-CoV2 and decreased levels of inflammatory response. A new section was included to discuss Nrf2 activation and COVID-19 infection (Please see section #4).

  1. Please cite the research article with references relevant to COVID-19 for NRF2 activation by contributing factors:

Response: We appreciate the reviewers' advice and included a new section to the review named: "Nrf2 activation and COVID-19 Infection".

We included new and recent references, as requested, and discussed the activation of Nrf2 signaling in COVID-19 patients. Although new publications are still emerging in this area, we found that pharmacological activation of Nrf2 may activate several cytoprotective genes and improve inflammatory response in COVID-19 patients. Lung biopsies from infected patients showed suppression of the Nrf2 pathway, but Nrf2 pharmacological inducers inhibited the replication of SARS-CoV2 and decreased levels of inflammatory response. Experimental evidence showed that NRF2 activators, such as dimethyl fumarate and 4-octyl itaconate, were able to suppress the inflammatory response to SARS-CoV2 in human cells, including peripheral blood mononuclear cells from COVID-19 patients (Please see section 4).

  1. In the sentence "Moreover, we provide a new approach to target ……………………….compounds in combination with vitamin D3" authors must modify the phrase to suggest the approach and tone down the sentence as they are not mentioning about their finding.

Response: As recommended, we modified the sentence, which now reads: "Moreover, we suggest a new approach..." (Conclusion - line 645).

  1. Likewise, they should modify other sentences with reference to SARS-CoV2 infection.

Response: The manuscript was revised according to the reviewer's advice.

  1. Few sentences need grammatical correction.

Overall, I recommend minor revision and precision in the review.

Response: As advised by the reviewer, the manuscript was revised for grammatical correction and precision in the review.

Round 2

Reviewer 3 Report

The revised manuscript by P. Mendonca and K.F.A. Soliman has been improved by the addition of information that were previously missing to the reader. This manuscript aims to present a review of the main biological involvements of the NFE2L2/ARE pathway and their relationship with SARS-CoV infection and this goal is mainly achieved. However, the link between NFE2L2/ARE pathway and SARS-CoV infection is mainly based on the extrapolation of disparate data (obtained from various types of viruses and various human or mouse data, cellular or animal models) and in that, this manuscript is more a hypothesis than a review.

In addition, some specific concerns remain to be addressed.

1) In paragraph 7, the authors explained a little bit about the link between epigenetic regulation, oxidative stress and NFE2L2, and I thank them for that. However, this paragraph mainly deals with epigenetic regulation and its title must be changed accordingly. Something like “oxidative stress-mediated hypomethylation increases expression of ACE2 receptor” would seem to better reflect the content of the paragraph.

2) Although some reorganization of the manuscript has been done with respect to the previous version, some redundancies remains between paragraphs. Some of these repetitions may be useful due to the length of the manuscript but the "copy and paste" of whole sentences is not acceptable. Two examples of this:

  • Lines 99-102, p.3: “Upon viral infection, redox-sensitive Keap1 cysteine residues (Cys151, Cys273, Cys288), which are very susceptible to theconjugation of a variety of reactive oxygen species (ROS)-inducing agents, get conjugated and the Nrf2 ubiquitination mediated by Keap1 is drastically decreased [29,30].”
  • Lines 103-105, p.3:"Nrf2 translocates to the nucleus and forms a complex with Maf proteins and other coactivators, binding to AREs in the promoter region, activating the transcription of cytoprotective and detoxifying genes [34-37].”

These two sentences are exactly the same as the following:

  • Lines 264-266, p.7:“Upon viral infection, redox-sensitive Keap1 cysteine residues, which are very susceptible to the conjugation of a variety of ROS-inducing agents, get conjugated and the Nfr2 ubiquitination mediated by Keap1 is drastically decreased [29,30].
  • Lines 266-268, p.7:“Nrf2 translocate to the nucleus and forms a complex with Maf proteins and other coactivators, binding to AREs in the promoter region, activating the transcription of cytoprotective and detoxifying genes [34-37].”

Author Response

Dear editor:

We are pleased to resubmit the revised version of the Manuscript ID: antioxidants-846929 with the modifications requested, including Title change: "Flavonoids Activation of the Transcription Factor Nrf2 as a Hypothesis Approach for Prevention and Modulation of SARS-CoV Infection Severity". We appreciate all the advice and requests, and we have addressed each concern as outlined below.

EDITOR

I agree with the major concern of one of the reviewers that this manuscript has to be clearly marked as a "hypothesis paper". Please include "hypothesis" and/or "hypothesis paper" both in the title and in the abstract. Also, the "Conclusions" section at the end of the manuscript should start with "In this hypothesis paper,..." instead of "In this review,...".

Response: The modifications were made to title, abstract (line 22), introduction (line 48), and conclusion (lines 622 and 643), as requested.

REVIEWER 3:

The revised manuscript by P. Mendonca and K.F.A. Soliman has been improved by the addition of information that was previously missing to the reader. This manuscript aims to present a review of the main biological involvements of the NFE2L2/ARE pathway and their relationship with SARS-CoV infection, and this goal is mainly achieved. However, the link between NFE2L2/ARE pathway and SARS-CoV infection is mainly based on the extrapolation of disparate data (obtained from various types of viruses and various human or mouse data, cellular or animal models) and in that, this manuscript is more a hypothesis than a review.

Response: As requested by the reviewer, we are referring to this manuscript as a hypothesis and not as a review anymore. Changes were made to the title, abstract (line 22), introduction (line 48), and conclusion (lines 622 and 643).

1) In paragraph 7, the authors explained a little bit about the link between epigenetic regulation, oxidative stress, and NFE2L2, and I thank them for that. However, this paragraph mainly deals with epigenetic regulation, and its title must be changed accordingly. Something like "oxidative stress-mediated hypomethylation increases expression of ACE2 receptor" would seem to better reflect the content of the paragraph.

Response: We accept the reviewer's advice, and the title of chapter 7 was modified to: "Oxidative stress-mediated hypomethylation increases expression of ACE2 receptor" (line 204).

2) Although some reorganization of the manuscript has been done with respect to the previous version, some redundancies remain between paragraphs. Some of these repetitions may be useful due to the length of the manuscript, but the "copy and paste" of whole sentences is not acceptable. Two examples of this:

  • Lines 99-102, p.3: "Upon viral infection, redox-sensitive Keap1 cysteine residues (Cys151, Cys273, Cys288), which are very susceptible to the conjugation of a variety of reactive oxygen species (ROS)-inducing agents, get conjugated and the Nrf2 ubiquitination mediated by Keap1 is drastically decreased [29,30]."

Response: As requested, the sentence above was re-written (lines 100-102).

  • Lines 103-105, p.3: "Nrf2 translocates to the nucleus and forms a complex with Maf proteins and other coactivators, binding to AREs in the promoter region, activating the transcription of cytoprotective and detoxifying genes [34-37]."

Response: As requested, the sentence above was re-written (lines 104-106).

These two sentences are exactly the same as the following:

  • Lines 264-266, p.7: "Upon viral infection, redox-sensitive Keap1 cysteine residues, which are very susceptible to the conjugation of a variety of ROS-inducing agents, get conjugated and the Nfr2 ubiquitination mediated by Keap1 is drastically decreased [29,30].

Response: This part was deleted from the manuscript since it was duplicated.

  • Lines 266-268, p.7: "Nrf2 translocate to the nucleus and forms a complex with Maf proteins and other coactivators, binding to AREs in the promoter region, activating the transcription of cytoprotective and detoxifying genes [34-37]."

Response: This part was deleted from the manuscript since it was duplicated.

Round 3

Reviewer 3 Report

In their new revised manuscript, P. Mendonca and K.F.A. Soliman have addressed all previous concerns and corrected some additional minor points. However, minor mistakes remain concerning gene names on lines 439-440.  

  • Acetyl CoA carboxylase is a heterotetramer of two distinct subunits encoded by ACACA and ACACB genes in addition to the homodimer of biotin carrier protein encoded by the BCP gene. Thus, referring to ACC as a gene name is wrong.
  • The gene name for fatty acid synthase is FASN in order to avoid confusion with the pro-apoptotic receptor Fas (encoded by the FAS gene).
  • According to the more recent nomenclature, Srebp1C is call SREBF1 and, anyway, encoded by the SREBF1 gene.
  • All other gene names should be italicized and in uppercase (for human genes) without greek letters: PPARG instead of Pparγ, FABP4, CEBPACEBPB.

Author Response

Reviewer 3

In their new revised manuscript, P. Mendonca and K.F.A. Soliman have addressed all previous concerns and corrected some additional minor points. However, minor mistakes remain concerning gene names on lines 439-440.  

  • Acetyl CoA carboxylase is a heterotetramer of two distinct subunits encoded by ACACA and ACACB genes in addition to the homodimer of biotin carrier protein encoded by the BCP Thus, referring to ACC as a gene name is wrong.
  • The gene name for fatty acid synthase is FASN in order to avoid confusion with the pro-apoptotic receptor Fas (encoded by the FAS gene).
  • According to the more recent nomenclature, Srebp1C is called SREBF1 and, anyway, encoded by the SREBF1 gene.
  • All other gene names should be italicized and in uppercase(for human genes) without Greek letters: PPARG instead of PparγFABP4CEBPA

Response: We appreciate all the corrections made by the reviewer. The gene nomenclature was revised as advised ( Section13, lines: 395- 397)